# Grounded Time-Series Anomaly Detection with Compact Multimodal Models and Preference Optimization

Werner Laemlin [1]   Svitlana Vyetrenko [2]

## Abstract

We study time-series anomaly detection (TSAD) with compact multimodal models that jointly perform anomaly interval localization and generate verifiable natural-language explanations. We introduce a staged V0–V3 framework that progressively increases input richness and output structure, enabling a 3B vision-language model to close the gap with GPT-5.2 on out-of-distribution benchmarks while producing citation-grounded explanations. A key limitation is that post-SFT explanations exhibit structured grounding errors that invalidate factual correctness under strict verification. We therefore adopt DPOP, a positive-anchoring extension, to preserve the likelihood of well-formed outputs during optimization, raising the fully-truthful explanation rate from 72.4% to 92.6% while maintaining localization performance. Our results show that explicit anchoring makes grounded preference optimization practical for reliable structured reasoning with compact multimodal models under strict output constraints.

## 1. Introduction

Time-series anomaly detection (TSAD) is critical in industrial monitoring, healthcare, and finance, where deviations from normal behavior must be identified *and* explained. While traditional approaches focus on detection accuracy, they offer limited interpretability (Sørbø & Ruocco, 2023; Bhattacharya et al., 2024). Recent multimodal LLMs reason over both numerical and visual signals (Zhou & Yu, 2025; Shen et al., 2025) but remain computationally prohibitive for edge deployment (Yao et al., 2024).

We investigate whether *compact* models can perform detection and explanation jointly. Prior work shows that small

[1]BETA, Université de Strasbourg, France [2]Outsampler. Correspondence to: Werner Laemlin <laemlin.w@gmail.com>.

*Accepted at the 2nd ICML Workshop on Foundation Models for Structured Data (FMSD), 2026. Non-archival workshop paper.*

models trained on high-quality data can match larger systems (Eldan & Li, 2023; Guo et al., 2025). We train a 3B vision-language model to predict anomalous intervals and produce structured explanations with machine-verifiable inline citations of the form *(feature@timestamp:value)*. This transforms explanation quality from a subjective assessment into a deterministic check against the input—a strict regime that surfaces failure modes invisible to plausibility metrics.

Our contributions are: (i) a staged V0–V3 framework that scales TSAD from pure interval detection to compact multimodal grounded reasoning; (ii) a distillation and verification pipeline for structured explanations with machine-checkable inline citations; and (iii) a grounding-oriented preference optimization setup, together with an empirical evaluation showing that it substantially improves explanation reliability under strict output constraints while preserving localization performance.

## 2. From Detection to Multimodal Grounded Reasoning

We formulate TSAD as structured prediction over $I = [S, E]$: predict $\hat{A} = [\hat{s}, \hat{e}]$ matching ground truth $A = [s, e]$. Point-wise protocols misleadingly credit any alarm inside a labeled window without penalizing poor temporal alignment (Bhattacharya et al., 2024). We instead adopt *affiliation metrics* (Huet et al., 2022), mapping directed distances to calibrated proximity scores in $[0.5, 1]$:

$$F_{\text{prec}}(d) = 1 - \frac{|A| + \min(d, s - S) + \min(d, E - e)}{|I|},$$
$$F_{\text{rec}}(d; y) = 1 - \frac{\min(d, y - S) + \min(d, E - y)}{|I|}. \quad (1)$$

Affiliation precision, recall, and $F_1$ are:

$$P_{\text{aff}} = \frac{1}{|\hat{A}|} \int_{x \in \hat{A}} F_{\text{prec}}(\text{dist}(x, A)) \, dx,$$
$$R_{\text{aff}} = \frac{1}{|A|} \int_{y \in A} F_{\text{rec}}(\text{dist}(y, \hat{A}); y) \, dy,$$
$$F1_{\text{aff}} = \frac{2 P_{\text{aff}} R_{\text{aff}}}{P_{\text{aff}} + R_{\text{aff}}}. \quad (2)$$

Unlike binary overlap, this rewards temporal proximity, penalizes both under- and over-estimation, and is calibrated across series of different lengths (0.5 = random, 1 = perfect).

We define four cumulative variants that separate changes in the output schema from changes in the evidence provided to the model.

**V0**: the model receives the raw time series as text and predicts only the anomalous interval.

**V1**: the observed time-series evidence remains limited to the raw signal, but the prompt and output schema are extended with a *description* field. This tests the effect of requiring an explanation without adding new input features.

**V2**: the output schema is unchanged from V1, while the textual input is augmented with a backward-looking moving average and moving standard deviation (window 30) (Park et al., 2026). This tests the value of explicit local trend and variability features without using future observations.

**V3**: the output schema remains unchanged, and each textual feature from V2 is additionally paired with its plot image. This tests whether visual representations add information beyond the textual signals (Xu et al., 2026; Zhou & Yu, 2025).

## 3. Training Objective

The model optimizes $\mathcal{L} = \mathcal{L}_{LM} + \lambda\mathcal{L}_{ITC}$, with $\lambda = 0.1$. $\mathcal{L}_{LM}$ is the standard next-token cross-entropy over the structured anomaly report. For V3, we add a CLIP-style InfoNCE term (Lu et al., 2019) that aligns an evidence embedding, pooled from the visual input tokens, with an answer embedding, pooled from the report tokens. In each batch, the matching evidence–answer pair is treated as positive, while the other pairs are used as negatives.

## 4. Distillation of Grounded Explanations

We train Qwen2.5-VL-3B-Instruct (Bai et al., 2025) via LoRA on GPT-4o teacher annotations. The synthetic set covers seven anomaly types: point, noisy-point, range, trend, noisy-trend, frequency, and noisy-frequency (Rousseau et al., 2025; Zhou & Yu, 2025). Because preliminary NLI-based checks were not consistent enough for reliable entailment/rejection decisions, we adopt a purely verifiable grounding setup inspired by FActScore (Min et al., 2023): each explanation must support its claims with inline citations *(feature@ts:value)* that can be checked directly against the input.

For each series, a template description is refined by GPT-4o in two steps: first inserting citations drawn strictly from the input, then improving readability while preserving citations verbatim. All citations are verified deterministically,

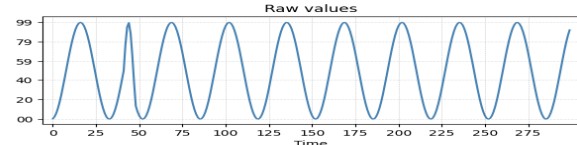

*(a)* Raw signal, true values of the time series.

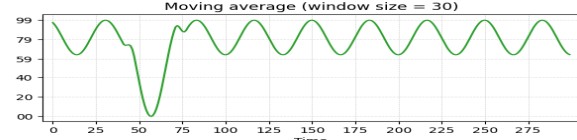

*(b)* Moving average (computed using a window size of 30).

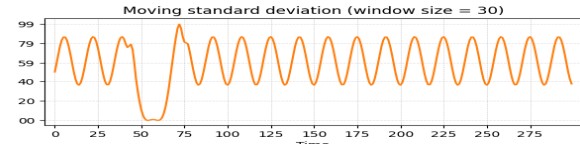

*(c)* Moving standard deviation (window size = 30).

*Figure 1.* Time series plots for a synthetic frequency increase anomaly over interval [42:49].

samples with any incorrect citation are discarded, yielding ≈500 fully grounded examples.

---

**INFERENCE PROMPT (V3 setup)**

Given the time series below and the following plot images: raw values, moving average, and moving standard deviation, determine whether there is an anomalous interval. Then, detect and describe the nature of the anomaly. Return ONLY a JSON object formatted exactly as follows, with no extra keys or text:

{ "anomalies": [
  { "start": int, "end": int, "description": string } ] }

Each description must be one short sentence explaining how the data deviates from normal. When mentioning specific data points, cite them using this format: (feature@index:value).

Time series:
timestamp: 1, value: 15, moving_avg: 02, moving_std: 09,
timestamp: 2, value: 17, moving_avg: 10, moving_std: 12,
timestamp: 3, value: 17, moving_avg: 23, moving_std: 15,
...

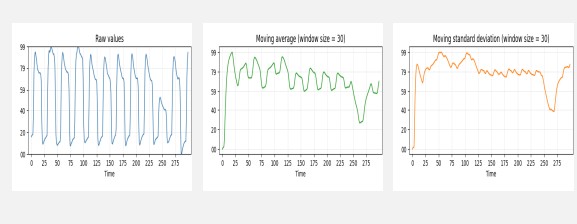

*Figure 2.* Inference prompt, example with maximum input richness, using UCR time series 174.

To address LLM tokenization failures on floats (Fons et al., 2024), all values are rescaled to two-digit integers (00–99) and formatted as explicit key-value pairs (e.g., *timestamp: 155, value: 36*) (Figure 2). Batches contain at most one sample per anomaly family, so negatives correspond to distinct anomaly mechanisms rather than near-duplicate examples.

## 5. Evaluation and Initial Results

We evaluate localization via affiliation metrics and explanation quality via *correct citation frequency* (fraction of citations matching the ground-truth series exactly) and *fully truthful explanation frequency* (all citations correct).

**Out-of-distribution (UCR).** After fine-tuning, V3 achieves $F_1 = 0.773$ on UCR (Dau et al., 2019), matching GPT-5.2 and outperforming GPT-4o (0.694) and XGBoost (0.680) (Table 1). The staged variants show that added evidence affects localization and grounding differently: V2 improves correct citations ($88.2\% \rightarrow 92.3\%$), while V3 recovers localization but remains limited in full truthfulness (72.4%). This structured factuality gap motivates the preference-optimization stage.

## 6. The Factuality Gap in Explanations

**Correct post-trained model output**

```
"anomalies": [{
    "start": 91,
    "end": 140,
    "description": "The data usually stays around
    a moving average of 62. Between indices 91 and
    140, it rises to 69 (moving_average@91:63,
    moving_average@140:67). During this period,
    the moving standard deviation, normally
    about 15, drops to 0 at index 138
    (moving_std@90:17, moving_std@138:0)."}]
```

*(a)* Correct output

**Post-trained model output with errors**

```
"anomalies": [{
    "start": 91,
    "end": 140,
    "description": "The data usually stays around
    a moving average of 62. Between indices 91 and
    140, it rises to 69 (value@91:63,
    value@140:67). During this period,
    the moving standard deviation, normally
    about 15, drops to 0 at index 136
    (moving_std@93:17, moving_std@136:0)."}]
```

*(b)* Output with feature confusion and time confusion examples.

*Figure 3.* Post-trained model outputs for anomaly detection.

**Temporal confusion**: the model retrieves a correct value but associates it with a nearby timestamp ($\leq \pm 4$ indices). **Feature confusion**: a value is correctly retrieved but attributed to the wrong feature. For V3 on UCR, these account for 32.1% and 17.9% of citation errors respectively.

These errors are especially damaging because citations are *explicitly auditable*: a single incorrect citation invalidates the full explanation. This reveals a gap between standard language modeling—optimizing fluency—and grounded reasoning, requiring per-claim factual precision against the input signal.

## 7. Preference Optimization under Structured Output Constraints

**Grounding-oriented preference construction.** For each prompt, the post-SFT model generates $n$ candidates. Those with any incorrect citation are discarded; the one with the highest affiliation $F_1$ serves as anchor $y_1$. A *better* candidate $y_0$ replaces the predicted interval with the ground truth while keeping the explanation unchanged. *Worse* candidates $y_2, y_3, \ldots$ inject one additional error per step—a temporal confusion (timestamp offset $\leq 4$) or a feature confusion (swap a cited feature name)—mirroring the dominant SFT failure modes. The ranked list is consumed as pairs $(y_0; y_2)$ by DPO-BT, or as a full ranking by DPO-PL.

**Unified listwise formulation.** Define $s_i = \beta \log(\pi_\theta(y_i|x)/\pi_{\text{ref}}(y_i|x))$ and let $\psi_i$ be the grounding quality score of candidate $y_i$. The general objective $\mathcal{L} = \mathbb{E}[l(\psi, s)]$ subsumes standard DPO objectives. With a Bradley–Terry model ($K = 2$):

$$\mathcal{L}_{\text{DPO}} = -\mathbb{E}[\log \sigma(s_w - s_l)]. \quad (3)$$

With full rankings, the Plackett–Luce likelihood uses the complete ordering:

$$\mathcal{L}_{\text{PL}} = -\mathbb{E}\left[\log \prod_{k=1}^{K} \frac{\exp(s_{\tau(k)})}{\sum_{j=k}^{K} \exp(s_{\tau(j)})}\right], \quad (4)$$

where $\tau$ denotes the permutation that orders candidates from best to worst according to their grounding quality.

**Structural degradation and the DPOP fix.** DPO caused formatting failures, often making outputs unparseable. Because preferred and rejected candidates differ by only a few citation tokens, preference optimization can increase relative margins while decreasing the absolute likelihood of well-formed outputs, as observed in Smaug-DPOP (Pal et al., 2024).

Following DPOP, we use a positive likelihood anchor, penalizing drops of the top-ranked grounded candidate below the reference model:

$$g^+(x, y) = \max\left(0, \log \frac{\pi_{\text{ref}}(y|x)}{\pi_\theta(y|x)}\right). \quad (5)$$

*Table 1.* Out-of-distribution UCR results: localization and citation quality for Qwen2.5-3B-VL variants and DPOP. Every value is averaged from 3 runs on the 180 time-series from the UCR dataset. XGBoost uses the current value and 10 backward lags for point-wise classification; the longest consecutive anomalous segment is used as its predicted interval.

| Metric | LLM-TSAD | GPT-4o | GPT-5.2 | XGBoost | Qwen2.5–3B-VL | | | | V3 + DPOP | | |
| --- | --- | --- | --- | --- | --- | --- | --- | --- | --- | --- | --- |
| | | | | | V0 | V1 | V2 | V3 | BT+ | PL+ | PL± |
| *Localization* | | | | | | | | | | | |
| Precision | .774 | .731 | .802 | .657 | .729 | .681 | .680 | .761 | .768 | .720 | .743 |
| Recall | .688 | .669 | .671 | .728 | .776 | .694 | .694 | .789 | .808 | .768 | .816 |
| $F_1$ | .724 | .694 | .773 | .680 | .742 | .677 | .677 | .773 | .775 | .732 | .765 |
| *Citation Quality* | | | | | | | | | | | |
| Correct citation freq. | – | .849 | .981 | – | – | .882 | .923 | .928 | .966 | .942 | .981 |
| Fully truthful expl. freq. | – | .784 | .975 | – | – | .728 | .728 | .724 | .885 | .794 | .926 |
| Avg. correct cit./expl. | – | 2.87 | 6.68 | – | – | 2.52 | 4.07 | 3.99 | 3.98 | 4.12 | 4.08 |
| Avg. incorrect cit./expl. | – | .51 | .13 | – | – | .34 | .34 | .31 | .14 | .26 | .08 |
| *Frequency of failure modes* | | | | | | | | | | | |
| Feature confusion/citation | – | .103 | .019 | – | – | .000 | .021 | .013 | .004 | .011 | .003 |
| Temporal confusion/citation | – | .030 | .000 | – | – | .061 | .026 | .023 | .010 | .018 | .004 |
| Other errors/citation | – | .018 | .000 | – | – | .057 | .030 | .036 | .020 | .028 | .012 |

PL± is our extension: it adds a negative anchor that penalizes probability increases for non-perfect candidates:

$$g^-(x, y) = \max\left(0, \log \frac{\pi_\theta(y|x)}{\pi_{\text{ref}}(y|x)}\right). \quad (6)$$

The anchored score is

$$\tilde{s}_i = s_i - \lambda_+ a_i g^+(x, y_i) - \lambda_- b_i g^-(x, y_i), \quad (7)$$

where $a_i = 1$ for the top-ranked grounded candidate and $b_i = 1$ for non-perfect candidates. We set $\lambda_+ = 20$, $\lambda_- = 1$. BT+ and PL+ use only the positive anchor ($b_i = 0$), whereas PL± uses both anchors. The underlying BT or PL objective is unchanged; only the candidate scores are anchored.

## 8. DPOP Results

Table 1 shows that DPOP improves grounding while largely preserving localization. Before DPO, the post-SFT V3 model reaches the same reported UCR localization $F_1$ as zero-shot GPT-5.2 ($F_1 = 0.773$), but only 72.4% of its explanations are fully truthful. BT+ raises this rate to 88.5% and slightly improves localization ($F_1 = 0.775$). The best overall trade-off is obtained by PL±, which reaches **92.6%** fully-truthful explanations and 98.1% citation correctness, matching GPT-5.2 on citation accuracy while maintaining strong localization ($F_1 = 0.765$).

The comparison between PL+ and PL± highlights the value of the negative anchor. Without it, PL+ provides only limited gains, reaching 79.4% fully-truthful explanations, 94.2% citation correctness, and $F_1 = 0.732$. With the negative anchor, PL± improves all three criteria: fully-truthful explanations rise to 92.6%, citation correctness reaches 98.1%, and localization improves to

$F_1 = 0.765$. It also reduces average incorrect citations from 0.26 to 0.08 per explanation.

These gains suggest that PL± benefits not only from listwise ranking, but also from controlling the absolute likelihood of correct and incorrect structured outputs. Since preferred and rejected candidates often differ by only a few citation tokens, purely relative optimization can increase margins while still making imperfect outputs more likely. The negative anchor counteracts this by discouraging probability increases for non-perfect candidates, making PL± the strongest variant when strict explanation factuality is the primary objective.

## 9. Conclusion

We present a unified framework for grounded time-series anomaly detection with compact multimodal models, combining staged knowledge distillation (V0–V3) with grounding-oriented preference optimization. Using GPT-4o annotations, a 3B vision-language model matches GPT-5.2 on UCR localization with V3, while DPOP-PL± raises the fully-truthful explanation rate to 92.6% and matches GPT-5.2 on citation correctness (98.1%). Beyond detection accuracy, our results show that compact models can generate verifiable, citation-grounded explanations under strict structured output constraints. In this setting, anchoring both positive and negative preference examples is key to improving factual grounding while preserving localization performance. These findings highlight a practical path toward reliable, interpretable, and computationally efficient TSAD. Future work will extend this framework to multi-anomaly and multivariate settings.

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

---

**Algorithm 1** Grounding-Oriented Preference List Construction

---

**Input:** prompt $x$, ground-truth interval $I^\star$, post-SFT model $\pi_{\text{SFT}}$, number of samples $N$, maximum rank $M$
**Output:** ordered preference list $\mathcal{L} = [y_{\text{best}}, y_1, y_2, \ldots, y_M]$
$\mathcal{C} \leftarrow \text{SampleCandidates}(\pi_{\text{SFT}}, x, N)$
$\mathcal{G} \leftarrow \emptyset$
**for** each candidate $y \in \mathcal{C}$ **do**
   **if** $\text{Parseable}(y)$ **and** $\text{CorrectCitations}(y, x)$ **then**
      $s_y \leftarrow \text{AffiliationF1}(\text{Interval}(y), I^\star)$
      $\mathcal{G} \leftarrow \mathcal{G} \cup \{(y, s_y)\}$
   **end if**
**end for**
**if** $\mathcal{G} = \emptyset$ **then**
   **return** $\emptyset$
**end if**
$y_1 \leftarrow \arg\max_{(y, s_y) \in \mathcal{G}} s_y$
$y_{\text{best}} \leftarrow \text{ReplaceInterval}(y_1, I^\star)$
$\mathcal{L} \leftarrow [y_{\text{best}}, y_1]$
**for** $m = 2$ **to** $M$ **do**
   $y_m \leftarrow \text{InjectCitationError}(y_{m-1})$
   $\mathcal{L} \leftarrow \mathcal{L} \,\|\, [y_m]$
**end for**
**return** $\mathcal{L}$

---

Here, $\text{CorrectCitations}(y, x)$ holds only if every citation in $y$ matches the corresponding feature, timestamp, and value in the input time series. $\text{ReplaceInterval}(y_1, I^\star)$ replaces only the predicted anomaly interval while leaving the explanation text unchanged. $\text{InjectCitationError}$ introduces one additional citation-level grounding error, either temporal or feature confusion.

## A. Hyperparameters

Our SFT variants use LoRA with:

- 500 time-series, balanced among the 7 synthetic anomaly types

- epochs = 10, 10, 15, and 20 for V0, V1, V2, and V3 respectively.

- rank = 8, $\alpha = 32$

- learning-rate = $2 \times 10^{-5}$

- dropout = 0.05

- gradient accumulation = 4

- batch size = 2

Our DPO variants use the following parameters:

- epochs = 6, 8 and 9 for $BT+$, $PL+$ and $PL\pm$ respectively.

- 150 time-series, balanced among the 7 synthetic anomaly types

- learning-rate = $2 \times 10^{-5}$

- DPO's beta = 0.8

- $\lambda_+ = 20$

- $\lambda_- = 1$

*Table 2.* In-distribution affiliation F1 on the synthetic benchmark. Student variants V1–V3 correspond to increasing input richness: raw values, raw values plus rolling statistics, and rolling statistics plus aligned plots.

| Type | LLM-TSAD | GPT-4o | GPT-5.2 | XGBoost | V0 | V1 | V2 | V3 |
|---|---|---|---|---|---|---|---|---|
| freq | .967 | .996 | .998 | .954 | .999 | .938 | .992 | .985 |
| point | .998 | .998 | .999 | .958 | .997 | .980 | .894 | .986 |
| range | .999 | .794 | .666 | .991 | .959 | .994 | .991 | .992 |
| trend | .578 | .656 | .622 | .973 | .976 | .947 | .963 | .943 |
| noisy-freq | .855 | .733 | .935 | .938 | .765 | .661 | .993 | .977 |
| noisy-point | .947 | .949 | .989 | .952 | .999 | .783 | .971 | .961 |
| noisy-trend | .451 | .844 | .373 | .967 | .974 | .918 | .941 | .962 |
| Overall | .854 | .850 | .807 | .964 | .945 | .926 | .980 | .975 |

*Table 3.* Out-of-distribution UCR time series: localization and citation quality for Qwen2.5-3B-VL variants and DPOP. Every value is averaged from 3 runs on the 180 time-series from the UCR dataset. $\pm$ denotes one sample standard deviation computed over all individual predictions across runs.

| Metric | Qwen2.5–3B-VL | | | | V3 + DPOP | | |
|---|---|---|---|---|---|---|---|
| | V0 | V1 | V2 | V3 | BT+ | PL+ | PL$\pm$ |
| *Localization* | | | | | | | |
| Precision | .729$\pm$.29 | .681$\pm$.31 | .680$\pm$.32 | .761$\pm$.28 | .768$\pm$.25 | .720$\pm$.29 | .743$\pm$.25 |
| Recall | .776$\pm$.24 | .694$\pm$.26 | .694$\pm$.27 | .789$\pm$.24 | .808$\pm$.20 | .768$\pm$.25 | .816$\pm$.23 |
| $F_1$ | .742$\pm$.26 | .677$\pm$.28 | .677$\pm$.28 | .773$\pm$.25 | .775$\pm$.21 | .732$\pm$.26 | .765$\pm$.22 |
| *Citation Quality* | | | | | | | |
| Correct citation freq. | – | .882 | .923 | .928 | .966 | .942 | .981 |
| Fully truthful expl. freq. | – | .728 | .728 | .724 | .885 | .794 | .926 |
| Avg. correct cit./expl. | – | 2.52 | 4.07 | 3.99 | 3.98 | 4.12 | 4.08 |
| Avg. incorrect cit./expl. | – | .34 | .34 | .31 | .14 | .26 | .08 |
| *Frequency of failure modes* | | | | | | | |
| Feature confusion/citation | – | .000 | .021 | .013 | .004 | .011 | .003 |
| Temporal confusion/citation | – | .061 | .026 | .023 | .010 | .018 | .004 |
| Other errors/citation | – | .057 | .030 | .036 | .020 | .028 | .012 |

# B. Synthetic anomaly types

## B.1. Point anomaly

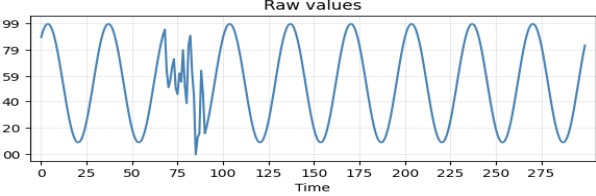

*(a)* Raw signal.

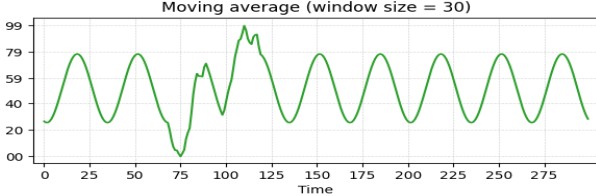

*(b)* Moving average.

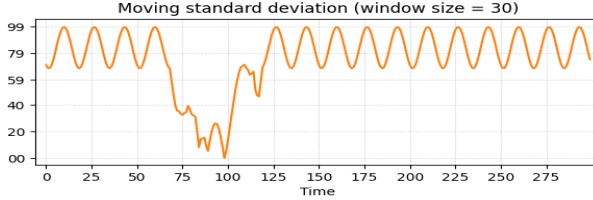

*(c)* Moving standard deviation.

*Figure 4.* Point anomaly: raw, moving average and standard deviation.

**Data generation parameters**

- Frequency: 0.03
- Normal duration rate: 240.0
- Anomaly duration rate: 25.0
- Minimum anomaly duration: 5
- Minimum normal duration: 100
- Anomaly standard deviation: 0.5

**Algorithmic description**

**V1**

Normally, the series is a noisy periodic sine wave centered on a stable level. From index 69 to 90, the time series shows extreme noise interference which disrupts the periodic pattern.

**V2-V3**

While the usual moving average is around 57, during the anomaly it drops down to a minimum of 20. The moving standard deviation is usually around 75, but at index 89 it drops down to 5.

**Annotation GPT-4o**

**V1**

The time series usually shows a smooth sine wave pattern. Between indices 69 and 90, there is a lot of noise, causing irregular changes. The value drops sharply from 95 to 64 (value@68:95, value@69:64). It continues to fluctuate, with a sudden drop to 0 at index 85 (value@85:00) and a rise to 64 at index 88 (value@88:64). This irregularity is different from the stable pattern seen before and after this section.

**V2-V3**

The time series usually shows a regular sine wave pattern. Between indices 69 and 90, there is a big disruption due to extreme noise. This noise changes the normal pattern.
During this anomaly:

- The moving average, normally around 47, drops to 0 (moving_average@68:26, moving_average@75:00).

- The moving standard deviation, usually around 86, falls to 5 at index 89 (moving_std@6:86, moving_std@89:05).

## B.2. Range anomaly

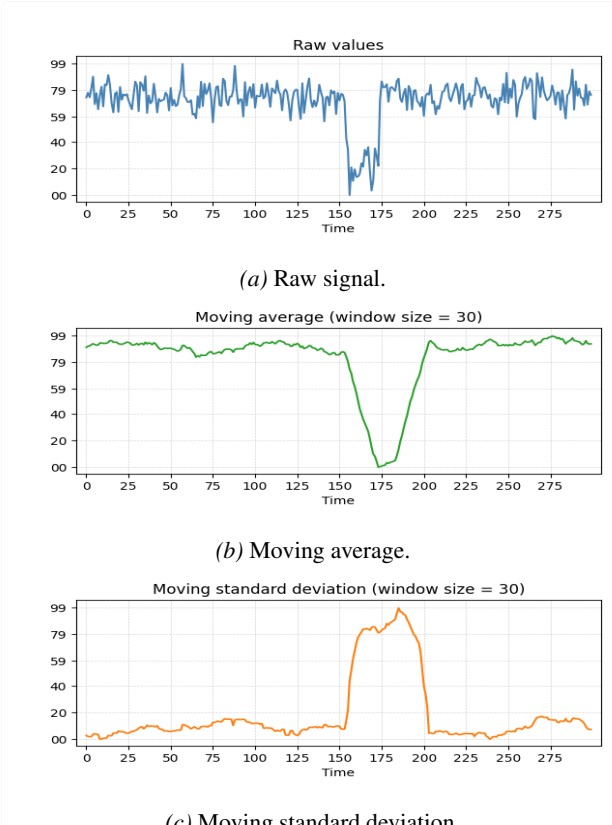

*(a)* Raw signal.

*(b)* Moving average.

*(c)* Moving standard deviation.

*Figure 5.* Range anomaly: raw, moving average and standard deviation.

**Algorithmic description**

**V1**

Normally, the series stays within a relatively narrow band. From index 154 to 174, the series exhibits a sustained downward level shift. While the usual moving average is around 73, during the anomaly it drops down to a minimum of 0. The moving standard deviation is usually around 9, but at index 170 it rises up to 85.

**V2-V3**

While the usual moving average is around 73, during the anomaly it drops down to a minimum of 0. The moving standard deviation is usually around 9, but at index 170 it rises up to 85.

**Annotation GPT-4o**

**V1**

The series usually stays within a narrow range. Notice the anomaly between indices 154 and 174. At index 154, the value is 43 (value@154:43). It drops sharply to 3 by index 169 (value@169:3). Then, it slightly recovers to 72 at index 174 (value@174:72). This drop is significant compared to the previous value of 71 at index 153 (value@153:71), highlighting the anomaly.

**V2-V3**

The time series is usually stable, but between indices 154 and 174, there is a clear drop. The moving average, normally around 73, falls to 0 (moving_average@153:86, moving_average@174:00). At the same time, the moving standard deviation, usually about 9, spikes to 85 at index 170 (moving_std@153:08, moving_std@170:85). This shows a big increase in variability during this period.

**Data generation parameters**

- Nominal standard deviation: 0.1
- Normal duration rate: 240.0
- Anomaly duration rate: 20.0
- Anomaly size range: [0.5, 0.8]
- Minimum anomaly duration: 5
- Minimum normal duration: 100

## B.3. Frequency anomaly

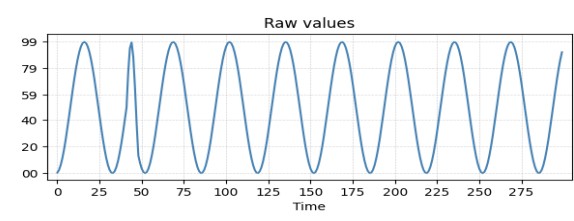

*(a)* Raw signal.

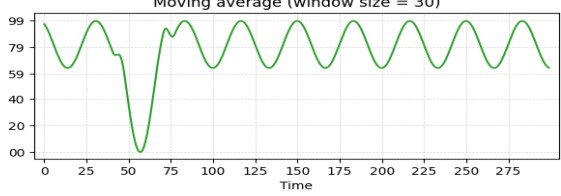

*(b)* Moving average.

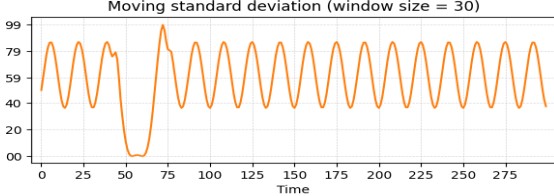

*(c)* Moving standard deviation.

*Figure 6.* Frequency anomaly: raw, moving average and standard deviation.

### Data generation parameters

- Frequency: 0.03
- Normal duration rate: 135.0
- Anomaly duration rate: 15.0
- Minimum anomaly duration: 7
- Minimum normal duration: 100
- Frequency multiplier: 3.0

### Algorithmic description

#### V1

Normally, the series has a stable oscillation frequency. From index 42 to 49, there is an anomaly characterized by a change in the signal's frequency: frequency is briefly multiplied by approximately 3.0 during the anomaly interval.

#### V2-V3

While the usual moving average is around 81, during the anomaly it goes down to a minimum of 00. The moving standard deviation is usually around 63, but at index 48 it drops down to 30.

### Annotation GPT-4o

#### V1

The series usually shows a steady oscillation pattern. Between indices 42 and 49, there is a noticeable anomaly. The frequency of oscillation increases about three times. Values change quickly here. For example, the value goes from 76 at index 42 to 94 at index 43, peaks at 99 at index 44, and drops to 65 by index 46 (value@42:76, value@43:94, value@44:99, value@46:65, value@49:08). This rapid change is different from the slower changes before and after this section.

#### V2-V3

The series usually has a stable oscillation frequency. Between indices 42 and 49, there is a clear anomaly. The frequency increases about threefold, with values fluctuating more rapidly (value@41:50, value@42:76). The moving average, normally around 74 before the anomaly (moving_average@41:74), rises sharply to 74 during the anomaly (moving_average@44:74). This shows a significant increase in the average signal level. The moving standard deviation, typically about 79 before the anomaly (moving_std@41:79), drops to 30 at index 48 (moving_std@48:30). This indicates reduced variability in the signal during the anomaly.

## B.4. Trend anomaly

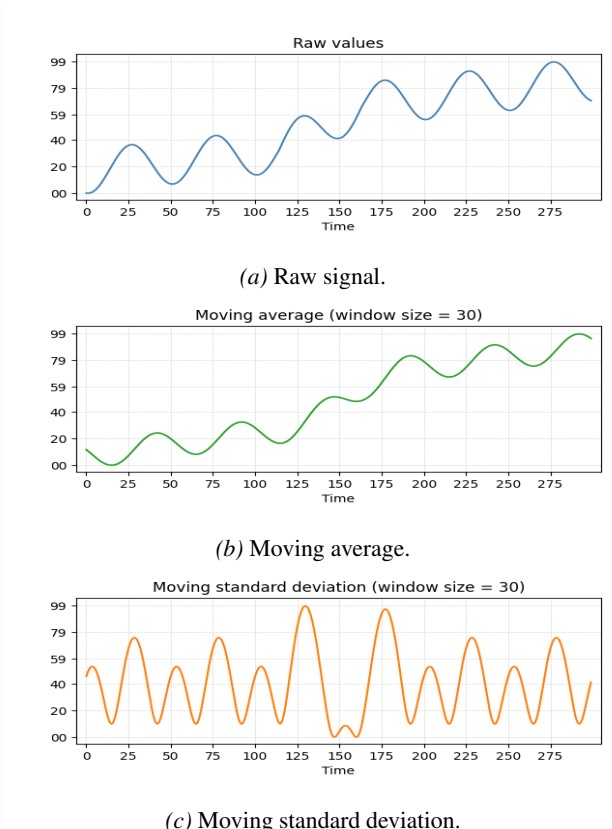

*(a)* Raw signal.

*(b)* Moving average.

*(c)* Moving standard deviation.

*Figure 7.* Trend anomaly: raw, moving average and standard deviation.

### Algorithmic description

#### V1

Normally, the series is a sine wave whose baseline drifts slowly. From index 114 to 164, there is an anomaly characterized by a trend-shift: baseline slope is multiplied by approximately 5.2 during the anomaly.

#### V2-V3

While the usual moving average is around 28, during the anomaly it rises up to a maximum of 65. The moving standard deviation is usually around 41, but at index 129 it rises up to 99.

### Annotation GPT-4o

#### V1

The series usually follows a sine wave with a slowly changing baseline. An anomaly occurs between indices 114 and 164. Here, the baseline rises sharply. The slope is about 5.2 times steeper than normal. Values increase quickly from 32 at index 114 to 67 at index 164 (value@114:32, value@164:67). Before and after this range, changes are slower.

#### V2-V3

During the anomaly period from index 114 to 164, there is a sharp change in trend. The baseline slope increases sharply by about 5.2 times. The moving average rises from around 28 (moving_average@100:28) to a peak of 65 (moving_average@164:49). The moving standard deviation, usually around 41 (moving_std@100:46), spikes to 99 at index 129 (moving_std@129:99). These changes show the anomaly's impact on trend and variability.

### Data generation parameters

- Frequency: 0.02
- Normal duration rate: 510.0
- Anomaly duration rate: 100.0
- Minimum anomaly duration: 50
- Minimum normal duration: 150
- Normal slope: 3.0
- Abnormal slope range: [6.0, 20.0]

## B.5. Noisy-point anomaly

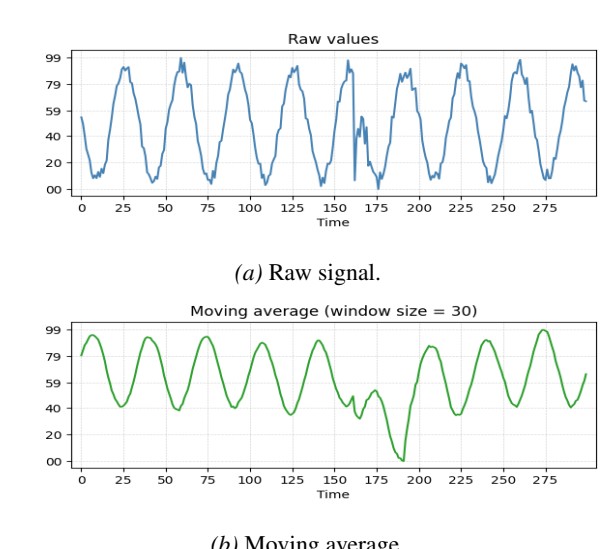

*(a)* Raw signal.

*(b)* Moving average.

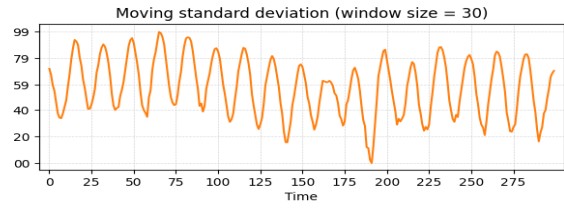

*(c)* Moving standard deviation.

*Figure 8.* Noisy-point anomaly: raw, moving average and standard deviation.

### Data generation parameters

- Frequency: 0.03
- Normal duration rate: 240.0
- Anomaly duration rate: 25.0
- Minimum anomaly duration: 5
- Minimum normal duration: 100
- Anomaly standard deviation: 0.5

### Algorithmic description

#### V1

Normally, the series is a noisy periodic sine wave centered on a stable level. From index 162 to 171, the time series shows extreme noise interference which briefly disrupts the periodic pattern.

#### V2-V3

While the usual moving average is around 45, during the anomaly it drops down to a minimum of 7. The moving standard deviation is usually around 48, but at index 162 it rises up to 62.

### Annotation GPT-4o

#### V1

The time series usually shows a noisy sine wave pattern around a stable level. An anomaly occurs between indices 162 and 171. At index 162, the value drops sharply to 7 (value@162:07). It then jumps to 38 at index 163 (value@163:38). The values continue to fluctuate, reaching 46 at index 164 (value@164:46) and dropping to 18 at index 170 (value@170:18). The pattern becomes stable again at index 171 with a value of 21 (value@171:21).

#### V2-V3

The time series usually shows a noisy sine wave pattern around a stable level. From index 162 to 171, noise increases significantly, disrupting this pattern. The moving average, normally around 45, drops to 33 (moving_average@161:49, moving_average@164:33). At the same time, the moving standard deviation, usually around 48, spikes to 62 at index 162 (moving_std@161:54, moving_std@162:62). This indicates unusual variability and deviation from the expected pattern.

## B.6. Noisy-trend anomaly

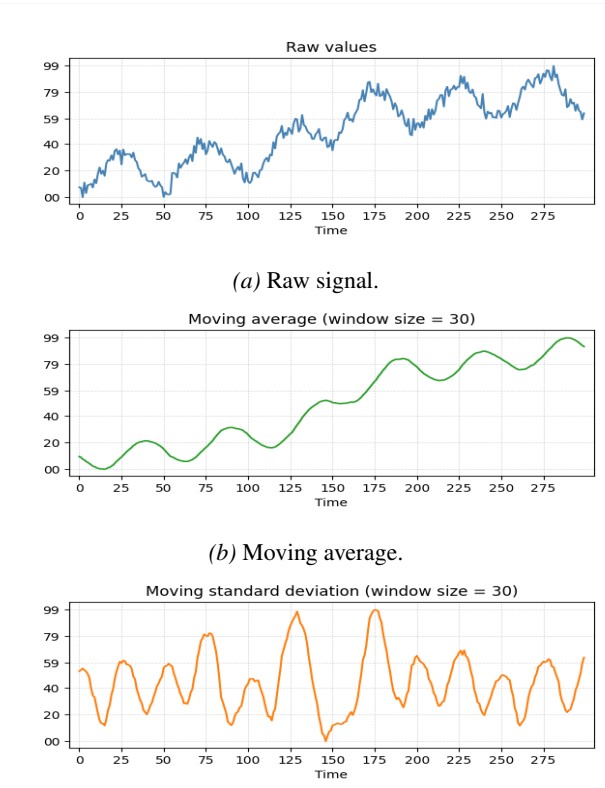

*(a)* Raw signal.

*(b)* Moving average.

*(c)* Moving standard deviation.

*Figure 9.* Noisy-trend anomaly: raw, moving average and standard deviation.

### Algorithmic description

#### V1

Normally, the series is a noisy sine wave whose baseline drifts slowly. From index 112 to 162, there is an anomaly characterized by a trend-shift: baseline slope is multiplied by approximately 5.6 during the anomaly.

#### V2-V3

While the usual moving average is around 27, during the anomaly it rises up to a maximum of 64. The moving standard deviation is usually around 40, but at index 129 it rises up to 98.

### Annotation GPT-4o

#### V1

The time series usually shows a noisy sine wave with a slowly changing baseline. An anomaly occurs between indices 112 and 162. Here, the baseline slope increases sharply, about 5.6 times. This is seen in the quick rise in values: from 27 at index 112 to 54 at index 120 (value@112:27, value@120:54). The trend continues with values reaching 59 at index 129 (value@129:59) and peaking at 64 by index 159 (value@159:64). This rise is much steeper than the gradual changes outside this range.

#### V2-V3

The time series usually shows a noisy sine wave with a slowly changing baseline. An anomaly occurs between indices 112 and 162, where the baseline slope increases sharply, about 5.6 times the usual rate. Normally, the moving average is around 27 (moving_average@100:26, moving_average@110:17), but during the anomaly, it rises to 64 (moving_average@168:56). The moving standard deviation, which is typically around 40 (moving_std@70:63, moving_std@100:43), spikes to 98 at index 129 (moving_std@129:98).

### Data generation parameters

- Frequency: 0.02
- Normal duration rate: 510.0
- Anomaly duration rate: 100.0
- Minimum anomaly duration: 50
- Minimum normal duration: 150
- Normal slope: 3.0
- Abnormal slope range: [6.0, 20.0]

## B.7. Noisy-frequency anomaly

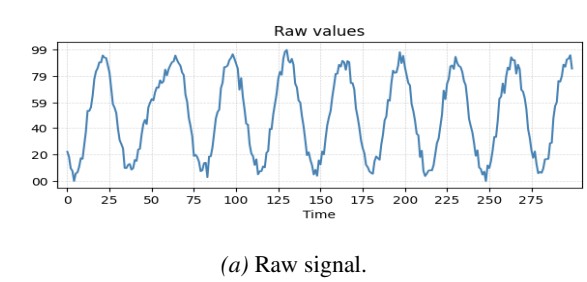

*(a)* Raw signal.

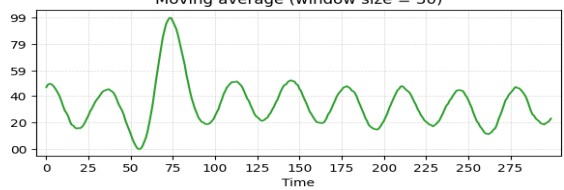

*(b)* Moving average.

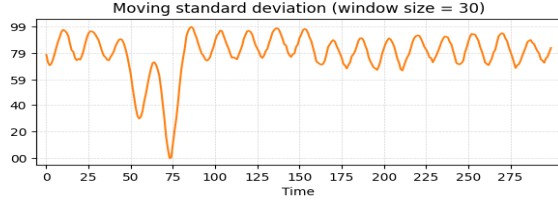

*(c)* Moving standard deviation.

*Figure 10.* Noisy-frequency anomaly: raw, moving average and standard deviation.

### Data generation parameters

- Frequency: 0.03
- Normal duration rate: 135.0
- Anomaly duration rate: 15.0
- Minimum anomaly duration: 7
- Minimum normal duration: 100
- Frequency multiplier: 3.0

### Algorithmic description

#### V1

Normally, the series is a noisy periodic sine wave centered on a stable level. From index 47 to 61, there is an anomaly characterized by a change in the signal's frequency: frequency is briefly divided by approximately 3.0 during the anomaly interval.

#### V2-V3

While the usual moving average is around 41, during the anomaly it rises up to a maximum of 84. The moving standard deviation is usually around 84, but at index 55 it drops down to 30.

### Annotation GPT-4o

#### V1

The time series usually shows a noisy sine wave pattern. An anomaly occurs between indices 47 and 61. Here, the wave frequency decreases significantly. Peaks and troughs are less frequent, with frequency about one-third of normal. Notice values at index 47 (value@47:43) and index 61 (value@61:87) show slower oscillation than usual.

#### V2-V3

The time series usually shows a noisy sine wave pattern around a stable level. An anomaly appears from index 47 to 61. Here, the frequency drops significantly, about three times slower. This means values change less often (value@46:46, value@54:70). The moving average, normally around 41, peaks at 84 during this anomaly (moving_average@47:21, moving_average@69:84). The moving standard deviation, usually around 84, drops to 30 at index 55 (moving_std@46:86, moving_std@55:30).

