# OpenReview forum: "Grounded Time-Series Anomaly Detection with Compact Multimodal Models and Preference Optimization"
_ICML.cc/2026/Workshop/FMSD — FMSD @ ICML 2026 Poster_

### Official Review · Reviewer_VFTR · 2026-05-21
**Grounded Time-Series Anomaly Detection with Compact Multimodal Models and Preference Optimization**

**Rating:** 5
**Confidence:** 5

**Review:**

## Summary
The paper studies whether a compact (3B) vision-language model can perform time-series anomaly detection (TSAD) jointly with grounded natural-language explanations whose factual claims are machine-verifiable against the input series. Two contributions: (i) a staged distillation pipeline V0→V3 that progressively enriches input richness (raw text, +description, +moving avg/std, +plot images) and output structure, with a verifier that drops samples containing any incorrect inline citation `feature@timestamp:value`; (ii) a grounding-oriented preference-optimization stage that builds preference lists from candidate samples ordered by affiliation-F1 grounding score, with citation-error-injected lower-ranked candidates, optimized via DPO-BT, PL+, and PL± variants, where PL± is a DPOP extension with both positive (likelihood floor) and negative (likelihood ceiling) anchoring. Trained via LoRA on Qwen2.5-VL-3B. On out-of-distribution UCR, V3 reaches localization F1=0.773, matching GPT-5 (0.773) and beating GPT-4o (0.694) and XGBoost (0.680); PL± raises fully-truthful explanations from V3's 72.4% to 92.6% and citation correctness to 98.1%, while preserving F1=0.765.

## Strengths
The grounding evaluation replaces subjective explanation-quality assessments with deterministic per-citation verification (`feature@timestamp:value` against the input), in the spirit of FActScore, which is appropriate for a domain where the input signal is fully observable and is portable to other multimodal-TSAD work. The staged enrichment V0→V3 decomposes input richness, output structure with citations, and optimization (SFT vs preference) along independent axes; Table 1 reads as a decomposition in which V2 raises citation correctness (88.2 → 92.3%) but localization stays flat, V3 lifts localization back to 0.773, and preference optimization closes the truthfulness gap. PL± targets the Smaug-DPOP failure mode in which DPO drives absolute likelihoods of well-formed outputs down because preferred and rejected candidates differ in only a few citation tokens, and the PL+ vs PL± comparison (79.4% → 92.6% truthful) is the direct evidence that absolute-likelihood anchoring is the operative mechanism. Affiliation metrics from Huet et al. 2022 are used for localization, sidestepping known critiques of point-wise overlap metrics. The preference-list construction (Alg. 1) specifies error injection, parseability checks, and top-1 replacement to ground truth, with hyperparameters in Appendix A.

## Weaknesses
The paper writing is messy and could be substantially tightened. Inputs and outputs of the model are not pinned down up front: which tokens, image tensors, and text fields enter the VLM, what the citation-tagged output string looks like as a formal object, and where the affiliation-F1 verifier sits in the pipeline are scattered across §3–§5 rather than fixed in a single notation block early on, which makes it hard to follow the rest of the paper as a coherent system. Relatedly, the DPOP / preference-optimization stage is presented after the results section rather than before, which inverts the natural reading order; the stage is also under-motivated, since the Smaug-style structural-degradation failure mode that DPOP / PL± targets (DPO driving absolute likelihoods of well-formed outputs down because preferred and rejected differ in only a few citation tokens) is not stated before the loss is introduced. The intuition behind PL± itself is not given: when the positive (likelihood floor) and negative (likelihood ceiling) anchors activate, what failure mode each one targets, and why the asymmetry λ_+ = 20 versus λ_- = 1 is the right setting are all left implicit, and a short conceptual paragraph plus a worked-example gradient on a near-miss versus a well-formed candidate would close this. A related-work section is missing entirely; given the page budget, the natural fix is to add one in the appendix that situates the contribution against prior multimodal TSAD work (TimeVLM and contemporaries), prior preference-optimization variants (SimPO, KTO, IPO, DPOP, R-DPO), and prior grounded-explanation evaluations (FActScore and downstream variants). The comparison against GPT-5 and GPT-4o is not like-for-like: the 3B Qwen2.5-VL is LoRA-finetuned with task-specific synthetic data, citation-token formatting, and a verifier-trained reward signal, while GPT-5 and GPT-4o are queried zero-shot through an API without prompt engineering, in-context exemplars, or a matched verifier scaffold; "matches GPT-5" therefore compares a task-tuned compact model against an untuned closed model, and a fairer comparison would either apply the same V0–V3 prompting + verifier scaffolding to GPT-5/GPT-4o or compare against GPT-5 with task-specific few-shot exemplars. Contemporary multimodal-TSAD baselines are also missing: TimeVLM is the natural reference point for a vision-language model on time-series tasks and is not run, and ChatTime, ChatTS, and SigLLM-style baselines are absent, so the F1 = 0.773 headline is anchored only against GPT-4o, GPT-5, and XGBoost. Beyond the new issues, only UCR is reported for OOD evaluation (with documented critiques in Wu & Keogh 2021), the seven synthetic anomaly families used in training are precisely the canonical UCR taxonomy so the OOD claim is generators-only rather than taxonomy-level, Table 1 averages three runs without reported variance leaving headline equalities such as V3 F1 = GPT-5 F1 = 0.773 unverifiable, teacher-student dependence on GPT-4o annotations is heavy, the preference-optimization comparison stays internal to the DPO family with no SimPO or KTO baseline, there is no human evaluation of explanation coherence (a citation-correct explanation that misinterprets the cited values would still pass the verifier), backbone scope is narrow (only Qwen2.5-VL-3B), and several reference entries contain literal "e. a." placeholders.

---

### Official Review · Reviewer_2saj · 2026-05-22
**Verifiable Explanations, Narrow Grounding Test**

**Rating:** 6
**Confidence:** 4

**Review:**

This is a focused and useful study of time-series anomaly detection as both localization and explanation. Its main strength is the evaluation protocol: explanations must cite concrete time-series values in a machine-checkable form, so factuality is not left to a reader’s impression. The staged V0-V3 design is also clear. Adding moving statistics and plot images separates the value of richer evidence from the value of merely asking for an explanation.

The strongest result is that the compact Qwen2.5-VL-3B setup reaches the reported GPT-5 localization F1 on UCR, while preference optimization sharply improves citation grounding. The DPOP PL± variant raises fully truthful explanations from 72.4% for V3 to 92.6%, with citation correctness reaching 98.1% and only a small localization trade-off.

The limits are important. The preference pairs are built around the same temporal and feature-confusion errors later measured, so the result may be partly a targeted repair rather than broad factual grounding. The evaluation is also still narrow: UCR is not a substitute for multivariate, multi-anomaly industrial data. Overall, the work is technically coherent and more rigorous than most explanation papers, but its generality remains unproven.

---

### Official Review · Reviewer_niAx · 2026-05-22
**Grounded Time-Series Anomaly Detection with Compact Multimodal Models and Preference Optimization**

**Rating:** 4
**Confidence:** 5

**Review:**

## Summary
This paper asks whether a small (3B parameter) vision-language model can perform time-series anomaly detection (TSAD) while simultaneously generating natural-language explanations whose factual claims are machine-verifiable against the input signal. The system is built in two phases. First, a staged knowledge-distillation pipeline (V0 through V3) progressively enriches both the input modality and the output structure: V0 consumes only textual time-series values and predicts anomalous intervals; V1 adds a description requirement; V2 augments with backward-looking moving average and standard deviation (window=30); V3 further includes rendered plot images (600x300 pixels). Throughout, explanations must contain inline citations of the form `feature@timestamp:value` that are deterministically verified against the input — any sample with even one incorrect citation is discarded during distillation, yielding ~500 fully-grounded training examples from GPT-4o teacher annotations. A CLIP-style InfoNCE term (lambda=0.1) aligns visual embeddings with report embeddings. Second, a grounding-oriented preference-optimization stage addresses the residual factuality gap: post-SFT V3 achieves F1=0.773 on UCR localization but only 72.4% fully-truthful explanations. A preference-list construction algorithm (Alg. 1) ranks candidate outputs by affiliation-F1, then injects citation errors (temporal confusion or feature confusion) into lower-ranked candidates. Three DPO variants are compared: BT (Bradley-Terry pairs), PL+ (Plackett-Luce with a positive likelihood-floor anchor), and PL± (adding a negative likelihood-ceiling anchor for imperfect candidates, with lambda_+=20, lambda_-=1). PL± raises fully-truthful explanations to 92.6% and citation correctness to 98.1% while maintaining localization at F1=0.765, matching GPT-5 on both detection and grounding metrics on out-of-distribution UCR evaluation.

## Strengths
- **Deterministic grounding evaluation.** Replacing subjective explanation-quality judgments with per-citation verification (`feature@timestamp:value` checked against the input) is a genuine methodological contribution. It mirrors FActScore-style atomic evaluation but exploits the fact that time-series inputs are fully observable, making verification exact rather than approximate. This is immediately reusable by the community.
- **Well-decomposed staged ablation (Table 1).** The V0-V3 progression isolates input richness from output structure from optimization strategy: V2 lifts citation correctness (88.2% to 92.3%) with flat localization, V3 recovers localization to 0.773 via visual evidence, and preference optimization closes the truthfulness gap. Each stage answers a clear sub-question.
- **PL± directly addresses the structural-degradation failure mode.** When preferred and rejected candidates differ by only a few citation tokens, standard DPO can increase relative margins while pushing absolute likelihoods of well-formed outputs below the reference model — the Smaug-DPOP phenomenon (Pal, 2024). The positive anchor prevents the model from "forgetting" correct formatting, while the negative anchor discourages increased probability on nearly-correct-but-flawed outputs. The PL+ vs PL± gap (79.4% to 92.6% truthful, 94.2% to 98.1% citation accuracy) is direct evidence that the negative anchor is the critical component.
- **Affiliation metrics (Huet et al. 2022) for localization.** Using directed-distance-based precision and recall instead of point-wise binary overlap sidesteps known issues with point-adjust and related metrics (Wu & Keogh 2021 critiques), and the metric is calibrated (0.5 = random, 1 = perfect) across different series lengths.
- **Practical value rescaling to 0-99 integers.** Addressing LLM tokenization failures on floating-point numbers (Fons, 2024) by rescaling all values to two-digit integers is a simple but impactful engineering choice that prevents a known failure mode from confounding the evaluation.
- **Detailed error taxonomy.** The paper quantifies failure modes: feature confusion (32.1% of citation errors on V3/UCR) vs. temporal confusion (17.9%) vs. other errors. This structured error analysis motivates the DPO stage more convincingly than a single aggregate metric would.
- **Algorithm 1 is fully specified.** The preference-list construction (sample N candidates, filter by parseability + citation correctness, rank by affiliation F1, replace top-1's interval with ground truth, inject one additional error per subsequent rank) is reproducible as stated.

## Weaknesses
- **Disorganized presentation.** The model's inputs and outputs are never pinned down in one place: what tokens, image tensors, and text fields enter the VLM, what the structured output string looks like formally, and where the verifier sits in the pipeline are scattered across Sections 3-5 rather than consolidated early. The preference-optimization stage appears after the results section, inverting the natural reading order. These structural choices make the paper harder to follow than the underlying ideas warrant.
- **Missing related-work section.** There is no dedicated related-work discussion. Prior multimodal TSAD work (TimeVLM, ChatTime, ChatTS, SigLLM), prior preference-optimization variants beyond DPOP (SimPO, KTO, IPO, R-DPO), and prior grounded-evaluation protocols (FActScore, SAFE) are not situated. Even a half-page appendix section would address this.
- **Unfair GPT-5 / GPT-4o comparison.** The 3B Qwen2.5-VL model is LoRA-finetuned on task-specific synthetic data with citation formatting, a verification-trained reward signal, and visual + textual inputs, while GPT-5 and GPT-4o are queried zero-shot via API. Claiming "matches GPT-5" under these asymmetric conditions is misleading. A fairer comparison would either (a) apply the same V3 prompting scaffold + verifier-filtered few-shot exemplars to GPT-5, or (b) clearly label this as "task-tuned compact model vs. general-purpose zero-shot large model" rather than implying parity.
- **No contemporary multimodal-TSAD baselines.** TimeVLM is the most natural reference point for a vision-language model on time-series tasks and is not evaluated. ChatTime, ChatTS, and SigLLM-style approaches are also absent. The F1=0.773 headline is anchored only against GPT-4o, GPT-5, and a simple XGBoost baseline, leaving the contribution's positioning against the actual TSAD-with-LLMs literature unclear.
- **OOD claim is overstated.** The seven synthetic anomaly families (point, noisy-point, range, trend, noisy-trend, frequency, noisy-frequency) used in training are precisely the canonical anomaly taxonomy that UCR itself uses. The "out-of-distribution" claim therefore covers generator-level novelty (different parameter draws) but not taxonomy-level novelty (unseen anomaly types). Real-world anomalies that don't fit these seven categories are untested.
- **No variance reporting on headline numbers.** Table 1 averages three runs but reports no standard deviation or confidence intervals. Key equalities (V3 F1 = GPT-5 F1 = 0.773) are unverifiable without dispersion estimates — the numbers could overlap within noise.
- **Heavy teacher dependence on GPT-4o.** The entire training pipeline starts from GPT-4o annotations; the 500 SFT examples are GPT-4o generated and filtered. If GPT-4o has systematic biases in how it describes certain anomaly types, those biases propagate. No analysis of teacher-error modes or robustness to teacher quality is provided.
- **Narrow backbone evaluation.** Only Qwen2.5-VL-3B is tested. Whether the approach transfers to other compact VLMs (LLaVA-1.5-7B, Phi-3-vision, InternVL2-2B) is unknown, leaving it unclear whether the gains are method-driven or backbone-lucky.
- **No human evaluation of explanation coherence.** A citation-correct explanation that misinterprets the cited values (e.g., correctly cites "moving_avg@91:63" but incorrectly concludes "the average is declining" when it's actually rising) would pass the verifier. Citation accuracy is necessary but not sufficient for explanation quality; even a lightweight human study on a subset would close this gap.
- **Preference-optimization comparison stays internal.** Only DPO-BT, PL+, and PL± are compared. SimPO and KTO — which don't require a reference model and might sidestep the structural-degradation issue differently — are not tested, leaving it unclear whether PL± is the best preference-optimization choice or merely the best within the DPOP family.
- **Incomplete references.** Several bibliography entries contain literal "e. a." placeholders instead of full author lists, suggesting the paper was rushed to submission.